# Multi-Band Frequency Window for Time-Frequency Fault Diagnosis of Induction Machines

**Jordi Burriel-Valencia, Ruben Puche-Panadero** **, Javier Martinez-Roman** ⓘ**,**
**Angel Sapena-Baño** \*ⓘ**, Martin Riera-Guasp** ⓘ **and Manuel Pineda-Sánchez** ⓘ

Institute for Energy Engineering, Universitat Politècnica de València, Camino de Vera s/n, 46022 Valencia, Spain
* Correspondence: asapena@die.upv.es; Tel.: +34-96-3877597

**Abstract:** Induction machines drive many industrial processes and their unexpected failure can cause heavy production losses. The analysis of the current spectrum can identify online the characteristic fault signatures at an early stage, avoiding unexpected breakdowns. Nevertheless, frequency domain analysis requires stable working conditions, which is not the case for wind generators, motors driving varying loads, and so forth. In these cases, an analysis in the time-frequency domain—such as a spectrogram—is required for detecting faults signatures. The spectrogram is built using the short time Fourier transform, but its resolution depends critically on the time window used to generate it—short windows provide good time resolution but poor frequency resolution, just the opposite than long windows. Therefore, the window must be adapted at each time to the shape of the expected fault harmonics, by highly skilled maintenance personnel. In this paper this problem is solved with the design of a new multi-band window, which generates simultaneously many different narrow-band current spectrograms and combines them into as single, high resolution one, without the need of manual adjustments. The proposed method is validated with the diagnosis of bar breakages during the start-up of a commercial induction motor.

**Keywords:** fault diagnosis; induction motors; wind energy generation; Fourier transforms; spectral analysis; spectrogram; transient regime

## 1. Introduction

Induction machines (IMs) are a key component of many industrial processes, either as motors or as generators, such as double fed induction generators (DFIGs) used for wind energy generation. Their reliability ensures the continuity of the production processes but they are subjected to eventual failures (broken bars, bearing faults, eccentricity, turn to turn or phase to ground short-circuits, etc.), which may cause unexpected breakdowns and high economic losses. A way to reduce these risks is the continuous monitoring of the machine's condition, in order to detect the presence of a fault at an early stage, when corrective measures can be implemented or maintenance works scheduled. Diverse quantities have been proposed in the technical literature for implementing condition based maintenance systems (CBMS) [1,2], such as the analysis of the stator currents [3–7], machine vibrations [8–10], fluxes [11,12], thermal images [13] or acoustic signals [14,15]. These techniques have been applied to detect different types of faults not only of the IM, such as stator inter-turn short circuits [11,16], broken bars [17,18], rotor asymmetries [19], eccentricity [20], bearing faults [6,21], but also of the inverter drive [3] or the mechanical coupling to the load, as gearboxes and pulleys [3,22].

A diagnostic technique that has gained a widespread interest in recent years is the motor current signature analysis (MCSA) [23–26], which is based on the detection of the characteristic fault signatures that each type of fault impresses in the current spectrum. It is a non-invasive method, which can be

applied on line without disturbing the normal operation of the machine; it requires, in its more basic implementation, just a current sensor for acquiring the current signal and a fast Fourier transform (FFT) for generating its spectrum; and it can detect a wide variety of machine faults through their spectral signatures, because each type of fault generate a specific set of fault frequencies, as shown in Table 1 [27], where $f_1$ stands for the fundamental component frequency, $p$ is the number of pole pairs, $s$ is the machine slip, $f_r$ is the rotational frequency of the IM rotor, $N_b$ is the number of balls of the bearings, $D_b$ if the bearing diameter, $D_c$ is the pitch or cage diameter and $\beta$ is the contact angle.

**Table 1.** Characteristic Frequencies of Different Types of Induction Machine (IM) Faults.

| Type of Fault | Fault Harmonics Frequency | |
|:---:|:---:|:---:|
| Shorted coils | $f_1\left(k \pm n\frac{1-s}{p}\right)$ | $k = 1, 3, 5\dots$<br>$n = 1, 2, 3, \dots$ |
| Rotor asymmetries | $f_1\left((1-s)\frac{k}{p} \pm s\right)$ | $\frac{k}{p} = 1, 3, 5\dots$ |
| Mixed eccentricity | $\|f_1 \pm k f_r\|$ | $k = 1, 2, 3\dots$ |
| Bearing (outer race) | $\frac{N_b}{2}f_r\left[1 - \frac{D_b\cos(\beta)}{D_c}\right]$ | |
| Bearing (inner race) | $\frac{N_b}{2}f_r\left[1 + \frac{D_b\cos(\beta)}{D_c}\right]$ | |
| Bearing (balls) | $\frac{D_c f_r}{2D_b}\left[1 - \left(\frac{D_b\cos(\beta)}{D_c}\right)^2\right]$ | |

Nevertheless, the use of the current spectrum as signal processing tool in MCSA limits its field of application to machines working in stationary conditions, which is not the case of industrial processes with varying load or speed conditions, or also of wind generators operating under variable wind regimes. In these cases, the fault frequencies shown in Table 1 become random time functions and the fault harmonics do not produce isolated spectral lines in the current spectrum, which blurs their characteristic signatures.

To extend MCSA to the fault diagnosis of IM working in transient regime, advanced time-frequency (TF) transforms of the current are needed, so that the transient fault signatures can be identified in a joint TF domain. These transforms can be linear, such as the short-time Fourier transform (STFT) [28–31], the short-frequency Fourier transform (SFFT) [31,32] and the wavelet transform (WT) [33], or quadratic, such as the Wigner-Ville distribution (WVD) [34] or the ambiguity function [8]. Quadratic TF transforms can achieve optimal resolution for mono-component chirp signals but, in case of multi-component ones, they produce cross-terms artifacts that pollute the TF representation of the current, making it difficult the correct identification of the fault harmonics. On the contrary, linear TF transforms are free from cross-terms artifacts. The STFT representation, the spectrogram, and the WVD representation are built by multiplying the current signal with an analysing window at each time instant and performing the FT of the resultant signal. In the case of the basic formulation of the STFT, this window has a constant shape, which makes it difficult to obtain a good resolution both in time and in frequency, since it uses a single window for all the parts of the TF-plane—a short window gives a good time resolution but a poor frequency resolution; on the contrary, a long window gives a good frequency resolution but a poor time resolution. The WT addresses this issue by performing a multi-resolution analysis, using different windows at different frequency bands—long windows for the lower frequency bands and short windows for the higher frequency bands. This strategy is a good trade-off in many applications but it does not suppose a significant advantage in the field of IM diagnosis, in which multi-component signals, with frequencies that can be very close, are handled. In this field the spectrogram must be constructed so that it is able to clearly separate harmonic components that evolve in overlapped frequency bands; this fact makes necessary to carefully select the shape of the window in every point of the TF-plane.

Diverse solutions to this problem have been proposed recently, in order to built a current's spectrogram with enough resolution for being used as an IM fault diagnosis tool. One of them is to adapt the shape of the analysing window in the TF domain to the expected shape of the fault harmonics, either using a single window, as in Reference [35] or using different window shapes for different sections of the current signal, as in References [36–38]. This approach requires a deep a priori knowledge of the time evolution of the fault harmonics in the current signal, which requires highly skilled maintenance personnel for implementing it and hinders its application in automated diagnostic systems. Other approach apply to the spectrogram a post-processing based on reassignment [39] or synchrosqueezing [9,40] techniques, so improving its sharpness, but these techniques add a considerable computational burden to the process of building the current spectrogram, departing from the simplicity of the STFT. Another alternative for obtaining an improved spectrogram is the Matching Pursuit approach [41,42]. This method is based on calculating a series of spectrograms, using a set of different windows designated as "dictionary" [43], which has to be previously built. Then, combining spectrograms corresponding to each window of the dictionary through a pre-defined algorithm, obtains the final spectrogram, which is considered the optimum one. This method has several drawbacks, such as the need of building an extensive dictionary, with a great amount of different windows for obtaining a good resolution spectrogram. This implies to calculate a huge quantity of spectrograms and thus consuming a vast quantity of time and computational resources

In this paper, a novel approach (up to the best of the authors' knowledge) is proposed to obtain a high-resolution current spectrogram, useful for fault diagnostic purposes, with the simplicity of a single STFT. It is based on

1. Performing the STFT with a wide range of windows with different lengths, and selecting, for each point in the TF domain, the best result obtained at that point among the complete set of windows.
2. Instead of running a separate STFT for each of the windows used in the analysis, a single, multi-band window is built by stacking all the desired analysing windows in consecutive frequency bands. This approach obtains in parallel the spectrograms corresponding to several hundreds of different analysis windows with the computing cost of a single one, which makes it suitable for fast, online diagnostic systems in transient regime.

The structure of the paper is as follows. In Section 2 the generation of the spectrogram with Gaussian windows shifted in the frequency domain is analyzed and in Section 3 it is used for the theoretical and practical explanation of the proposed method. In Section 4, it is validated with the analysis of the start-up current of a high rated power, medium voltage squirrel cage induction motor, with broken bars. Section 5 presents the conclusions of this work.

## 2. Time-Frequency Analysis of the Machine's Current via STFT with a Multi-Band Window

To highlight the spectral content of a time-varying current signal, which may contain the characteristic fault harmonics given in Table 1, it is necessary to generate a representation of the current signal in the join TF domain. Among the diverse transforms available to this end (STFT, SFFT, WT, WVD, ambiguity function, etc.), the STFT has been selected in this work, because it is a linear transform, without cross-terms artifacts, and is computationally very effective, which makes it suitable for being implemented in low-cost, low-power embedded devices for on-line CBM systems.

In this Section, the traditional STFT analysis of the stator current using a Gaussian window will be first reviewed and, after, the proposed method using a multi-band frequency window will be presented and compared with the traditional one.

### 2.1. Spectrogram of Machine's Current

The STFT of a current signal $i(t)$ is a linear TF transform that is able to generate a joint TF representation of the current, the spectrogram, through the following steps:

1. For each instant $\tau$, the current signal is multiplied element by element by the conjugate of a suitable time window centered at $\tau$, $h(t - \tau)$

$$i_\tau(t) = i(t)h(t - \tau)^* \tag{1}$$

   that emphasizes the content of the current signal at time $\tau$ and attenuates it at other times

$$i_\tau(t) = \begin{cases} i(t), & \text{if } t \text{ is close to } \tau \\ 0, & \text{if } t \text{ is far from } \tau \end{cases} \tag{2}$$

2. The Fourier transform if applied to the time-windowed signal $i_\tau(t)$, which gives the frequency content of the current signal $i(t)$ around time $\tau$

$$\begin{aligned} I_\tau(\omega) &= \tfrac{1}{\sqrt{2\pi}} \int e^{-j\omega t} i_\tau(t) dt \\ &= \tfrac{1}{\sqrt{2\pi}} \int e^{-j\omega t} i(t)h(t - \tau)^* dt \end{aligned} \tag{3}$$

   where $\omega = 2\pi f$ and $f$ stands for the frequency, in Hz.

3. The energy density spectrum at time $\tau$ is obtained as

$$I_{SP}(\tau, \omega) = |I_\tau(\omega)|^2 = \left| \frac{1}{\sqrt{2\pi}} \int e^{-j\omega t} i(t)h(t - \tau)^* dt \right|^2 \tag{4}$$

For each instant $\tau$ the STFT generates a different energy spectral density $I_{SP}(\tau, \omega)$ and the total set of these spectra constitutes the current spectrogram. A critical issue for obtaining a high resolution spectrogram of the current signal is the selection of the window $h(t)$ in (4). To obtain a high resolution of the energy content of the current signal in the joint TF domain, it is necessary to use a window with a high concentration of energy in the TF plane; but such energy concentration is limited by the Heisenberg-Gabor uncertainty principle —a short time window gives a good time resolution but a poor frequency resolution and, on the contrary, a long time window gives a good frequency resolution but a poor time resolution. The window that can achieve the highest energy concentration in the joint TF domain is the Gaussian window [35], given in the time domain by

$$g(t) = \left( \frac{\alpha}{\pi} \right)^{1/4} e^{-\frac{\alpha}{2}t^2} \tag{5}$$

and in the frequency domain by

$$G(\omega) = \left( \frac{1}{\alpha\pi} \right)^{1/4} e^{-\frac{1}{2\alpha}\omega^2} \tag{6}$$

The standard deviation of the Gaussian window in the time domain (5) is $\sigma_t^2 = 1/(2\alpha)$ and in the frequency domain (6) is $\sigma_\omega^2 = \alpha/2$. Therefore, for the Gaussian window, the product of its duration $\sigma_t$ and its bandwidth $\sigma_\omega$ gives [44]

$$\sigma_t \sigma_\omega = 1/2 \tag{7}$$

In the signal analysis field, the Heisenberg-Gabor uncertainty principle principle states that one cannot construct any signal whose duration $\sigma_t$ and bandwidth $\sigma_\omega$ are, both, arbitrarily small, because

$$\sigma_t \sigma_\omega \geq 1/2 \tag{8}$$

In this way, the Gaussian window has a duration-bandwidth product (7) that reaches the minimum value (i.e., the highest concentration in the joint TF plane) that can be achieved under the uncertainty principle (8).

The parameter $\alpha$ in (5) and in (6) is the only one that defines the shape of the Gaussian window. A low value of $\alpha$ gives a long window with a narrow bandwidth, while a high value of $\alpha$ gives a short window, with a wide bandwidth. This parameter must be tuned to the current signal to be analysed with the STFT. As detailed in Reference [35], the optimal Gaussian window to build the spectrogram of a given current signal is the one that has the maximum overlap with the current in the TF domain. That is, the optimal parameter $\alpha$ is the one whose height/width ratio ($\sigma_\omega/\sigma_t = \alpha$) best approximates the slope of the current signal in the TF domain. Unfortunately, the slope of the fault harmonics in an IM in transient regime is not a constant value, because in this regime the slip and the rotational frequency in Table 1 are time-varying quantities, as well as the fault frequencies that depend on them. Besides, different components of the current signal may have different slopes (such as the fundamental component and the fault harmonics). These facts preclude the use of a single, optimal Gaussian windows for building a high resolution diagnostic spectrogram of the IM current.

### 2.2. Frequency Shifting of the Gaussian Analysing Window

If the Gaussian window (5) is used as the analysing window to build the energy density spectrum, then (4) becomes

$$I_{SP}(\tau,\omega) = |I_\tau(\omega)|^2 = \left| \frac{1}{\sqrt{2\pi}} \int e^{-j\omega t} i(t) g(t-\tau)^* dt \right|^2 \tag{9}$$

The Gaussian window (5) is a real valued function. If it shifted in the frequency domain by a frequency $f_k$, corresponding to an angular frequency $\omega_k = 2\pi f_k$, then (5) becomes a complex-valued function

$$g_k(t) = g(t) e^{j\omega_k} = \left(\frac{\alpha}{\pi}\right)^{1/4} e^{-\frac{\alpha}{2}t^2} e^{j\omega_k} \tag{10}$$

Replacing (10) in (9) gives

$$I_{SP}(\tau,\omega) = |I_\tau(\omega)|^2 = \left| \frac{1}{\sqrt{2\pi}} \int e^{-j\omega t} i(t) g(t-\tau)^* e^{-j\omega_k} dt \right|^2 \tag{11}$$

that is,

$$I_{SP}(\tau,\omega) = |I_\tau(\omega)|^2 = \left| \frac{1}{\sqrt{2\pi}} \int e^{-j(\omega+\omega_k)t} i(t) g(t-\tau)^* dt \right|^2 \tag{12}$$

and, making the change of variable $\omega' = \omega + \omega_k$, (12) can be expressed as

$$I_{SP}(\tau,\omega'-\omega_k) = |I_\tau(\omega'-\omega_k)|^2 = \left| \frac{1}{\sqrt{2\pi}} \int e^{-j\omega' t} i(t) g(t-\tau)^* dt \right|^2 \tag{13}$$

Therefore, as the shift $\omega_k$ of the Gaussian analysis window changes, the frequencies of the entire corresponding spectrogram (13) suffer the same $\omega_k$ shift, at each time $\tau$.

### 3. Proposed Multi-Band Analysing Window

A possible approach for obtaining a high-resolution spectrogram of a current signal with components of different slopes in the TF domain could be a variant of the Matching Pursuit approach, using as dictionary a set of Gaussian functions with different values of $\alpha$. In this way, a batch of spectrograms, each one with a different value $\alpha$ for the Gaussian window (5) would be generated.

In a second stage, for each point in the TF domain, the best value obtained among the whole set of spectrograms would be selected, giving the best approximation to the ideal TF representation of the current signal.

The drawback of this technique is the high amount of resources that it requires. For each possible value of $\alpha$, in a given range, a full spectrogram must be built, which is a time-consuming operation and, also, it must be stored, which implies high memory resources. Afterwards, a processing algorithm must be applied to each point of all the spectrograms to combine them. These requirements make this technique unsuitable for being deployed with on-line, low power embedded devices.

On the contrary, the novel technique presented in this paper can achieve the same results at roughly the cost of a single STFT, in terms of speed and storage requirements, even with the use of several hundreds of Gaussian windows with different values of $\alpha$. It is based on a particular feature of the IM faults presented in Table 1—in most industrial IMs, the fault harmonics with the highest amplitudes are those with an index $k = 1$ and they are located in a narrow, low frequency band with a bandwidth $f_b$ (normally of one or two hundreds of Hz). Nevertheless, the current signal is acquired normally using high frequencies rates, from 5 or 10 kHz up to 100 kHz and more. This implies that, when performing the FT of the windowed current signal, at each stage of the STFT process, only the values within the narrow band $[0 - f_b]$ of diagnostic interest are kept and the rest of the spectrogram values are discarded, what represents a waste of computing resources.

The proposed method addresses this problem, filling the whole spectrogram with useful diagnostic contents. It relies on frequency shifting the Gaussian analysing window, as in (13). If the current signal is low-pass filtered with a cut-off frequency $f_b$ (using, for example, a frequency filter as in Reference [45]), then its spectrogram, built with a Gaussian window (5), will be non-zero only in the frequency band of diagnostic interest $[0 - f_b]$. But, if the Gaussian window is shifted to a frequency $f_k = f_b$ in (10), then the spectrogram of filtered current signal will appear in the frequency band $[f_b - 2f_b]$ (and will be zero outside this band). If the $\alpha$ parameters of both Gaussian windows (the shifted and the non-shifted one) are equal, the same spectral information will appear in both frequency bands. But, if the shifted Gaussian window has a different value of $\alpha$, then the spectrograms in the frequency bands $[0 - f_b]$ and $[f_b - 2f_b]$ will be different, each one corresponding to a different value of the parameter $\alpha$.

What is proposed in this paper is to extend this feature, using, instead of two frequency bands, a partition of the whole spectrogram in $N_g$ adjacent bands, each one with a frequency width equal to $f_b$; and to use a set of $N_g$ Gaussian windows, with different values of $\alpha$ ($\alpha_k$, $k = 0 \ldots N_g - 1$) and with increasing shifting frequencies ($f_k = kf_b$, $k = 0 \ldots N_g - 1$), for filling each of these frequency bands. This can be achieved in a single STFT execution (since in practice all the signals processed are discrete) if, using the superposition principle, all these $N_g$, frequency shifted, Gaussian windows are summed in the time domain, giving a single time window. The use of this new, multi-band window, in (4) gives a single spectrogram with $N_g$ adjacent bands, each one corresponding to the analysis of the current signal with a different parameter $\alpha_k$ for the Gaussian window.

For a given sampling frequency $f_s$ and a frequency band of interest $f_b$, the total number $N_g$ of adjacent frequency bands that can be used is equal to

$$N_g = f_s / f_b \tag{14}$$

The proposed multi-band window is built, using (10), as

$$gm(t) = \sum_{k=0}^{N_g-1} \left(\frac{\alpha_k}{\pi}\right)^{1/4} e^{-\frac{\alpha_k}{2}t^2} e^{jk2\pi f_b} \tag{15}$$

As an example, a multi-band window (15) has been built using $N_g = 10$ different Gaussian windows with values of $\alpha$ in (5) spanning linearly the range $[\alpha_{min} = 1 - \alpha_{max} = 700]$, with a step equal to $\Delta\alpha = (\alpha_{max} - \alpha_{min})/(N_g - 1)$, as

$$gm(t) = \sum_{k=0}^{N_g-1} \left(\frac{\alpha_{min} + k\Delta\alpha}{\pi}\right)^{1/4} e^{-\frac{\alpha_{min}+k\Delta\alpha}{2}t^2} e^{jk2\pi f_b} \tag{16}$$

The multi-band window (16) is displayed in Figure 1, which shows the real (Figure 1 top) and the imaginary part (Figure 1 middle) of the window, as well as its spectrum (Figure 1 bottom). Figure 2 shows the spectrogram of the window in the joint TF domain, with all the individual Gaussian atoms stacked in adjacent frequency bands.

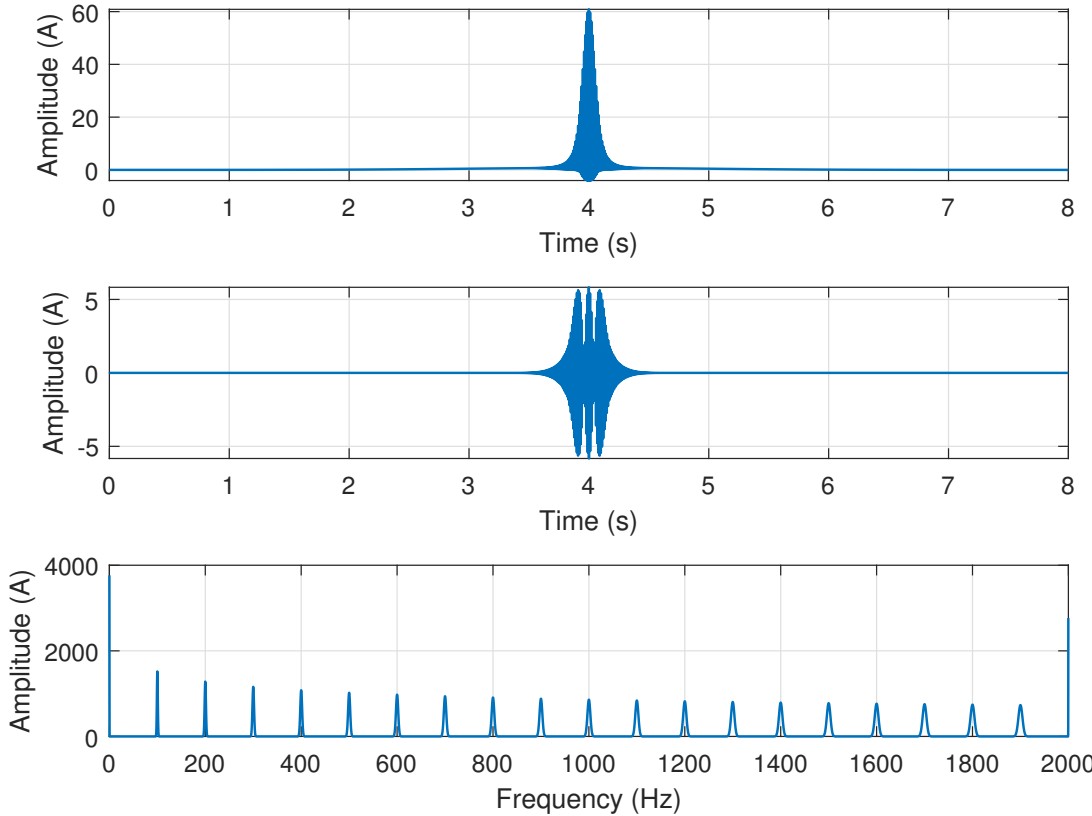

**Figure 1.** Multi-band window designed in Section 3 in the time domain (real part, top, imaginary part, middle) and in the frequency domain (bottom). This window contains 20 Gaussian windows, located in 20 adjacent frequency bands of 100 Hz, spanning linearly the range $[\alpha = 1 - \alpha = 700]$.

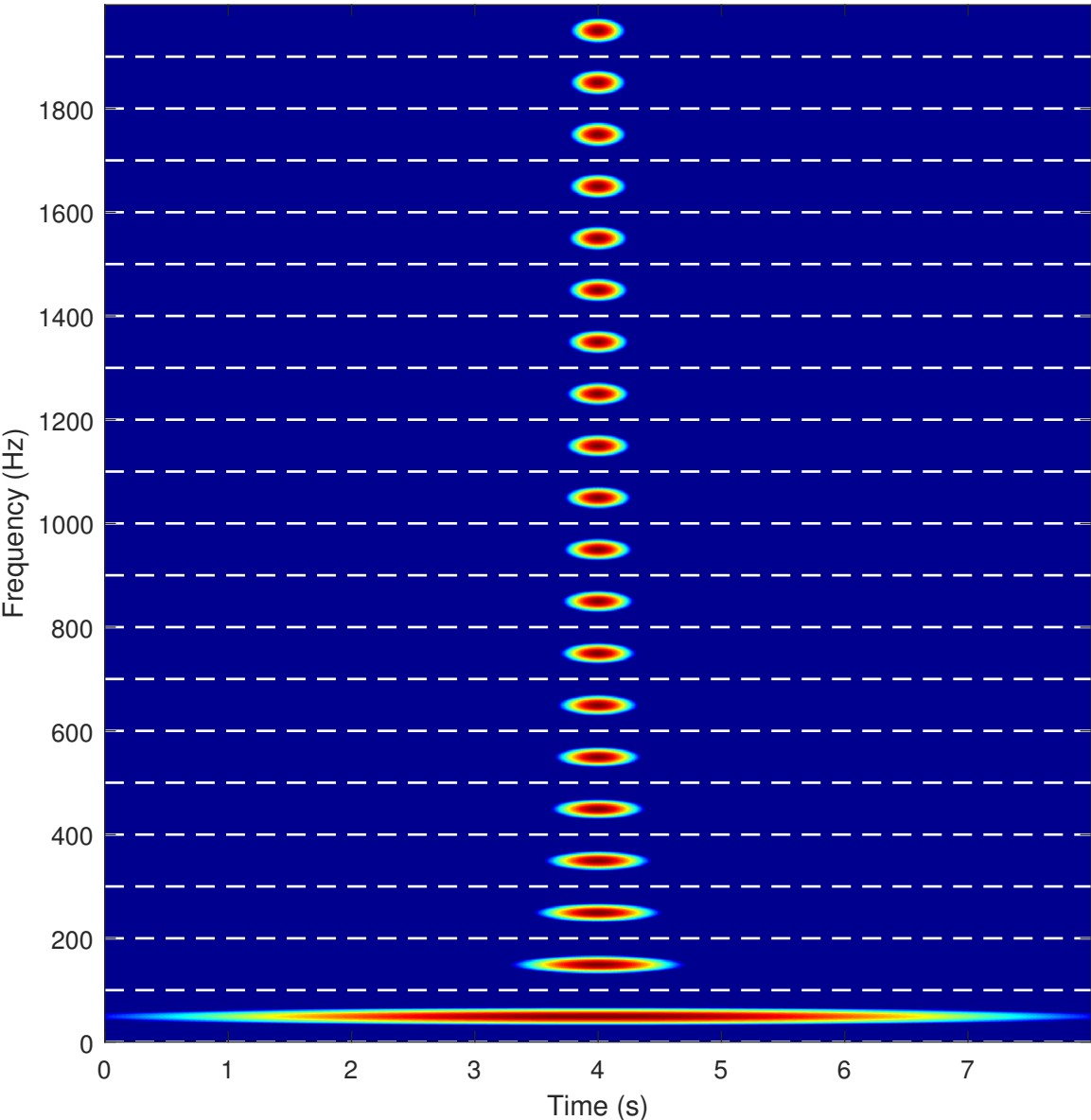

**Figure 2.** Atoms of the multi-band window designed in Section 3 in the joint time-frequency domain. This window contains 20 Gaussian windows, located in 20 adjacent frequency bands of 100 Hz, spanning linearly the range from $\alpha = 1$ (bottom) up to $\alpha = 700$ (top).

### 3.1. Steps for Applying the Proposed Multi-Band Window

In this sub-section the process for applying the concept of multi-band window to a given current signal $i(t)$ is explained, using a synthetic current signal $i(t)$ with a sinusoidal component of 50 Hz and a linear chirp with a frequency slope of 1 Hz/s, starting at $-50$ Hz. It has been built with a duration of 50 s, using a frequency sampling rate of 200 Hz (Figure 3),

$$i(t) = \cos(2\pi 50t) + 0.05\cos(2\pi(-50t + t^2)) \tag{17}$$

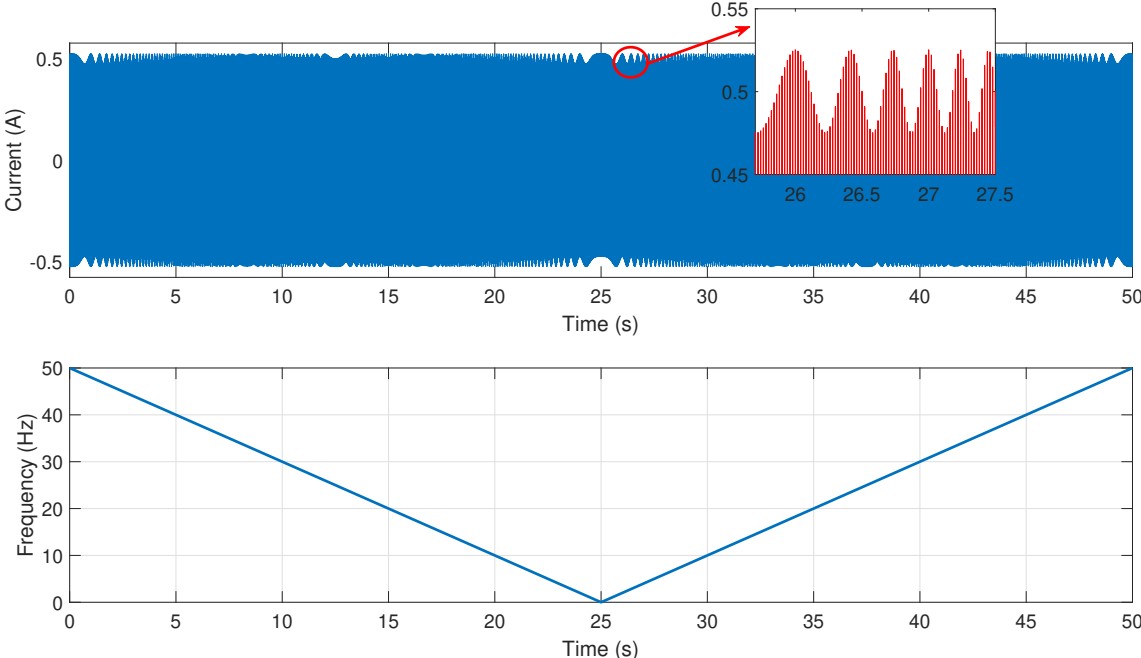

**Figure 3.** Synthetic current signal (17) used for illustrating the application of the proposed method (**top**). Frequency of the linear chirp component in the synthetic current signal (17) (**bottom**).

The steps for analysing the current signal (17) using the proposed multi-band windows are the following ones:

1.  The analysis window (15) is built:

    - first, the bandwidth of diagnostic interest $[0–f_b]$ is established ([0–100 Hz] in this case), which gives the maximum number of elementary Gaussian windows from (14) ($Ng = 200/100 = 2$). The current signal is low-pass filtered with a cut-off frequency equal to $f_b$. In this work, a spectral filter which zeroes all the spectrum bins with a frequency greater than $f_b$ has been used, as in Reference [45].
    - second, the parameters $\alpha_k$ of each of these windows in (15) must be selected. For an actual application of the method, a high number of $\alpha_k$ parameters would be automatically generated, as was explained in Section 3, applying an expression similar to (22) in Section 4. For the simple signal (17), only two values of $\alpha$ are used, $\alpha_0 = 1$ (a long window) and $\alpha_1 = 12.6$, tailored to the chirp component according to Reference [35].

The resulting multi-band window expression, applying (15), is

$$gm(t) = \left(\frac{1}{\pi}\right)^{1/4} e^{-t^2/2} + \left(\frac{12.6}{\pi}\right)^{1/4} e^{-12.6t^2/2} e^{j2\pi100t} \qquad (18)$$

This window is plotted in Figure 4, in the time and frequency domains.

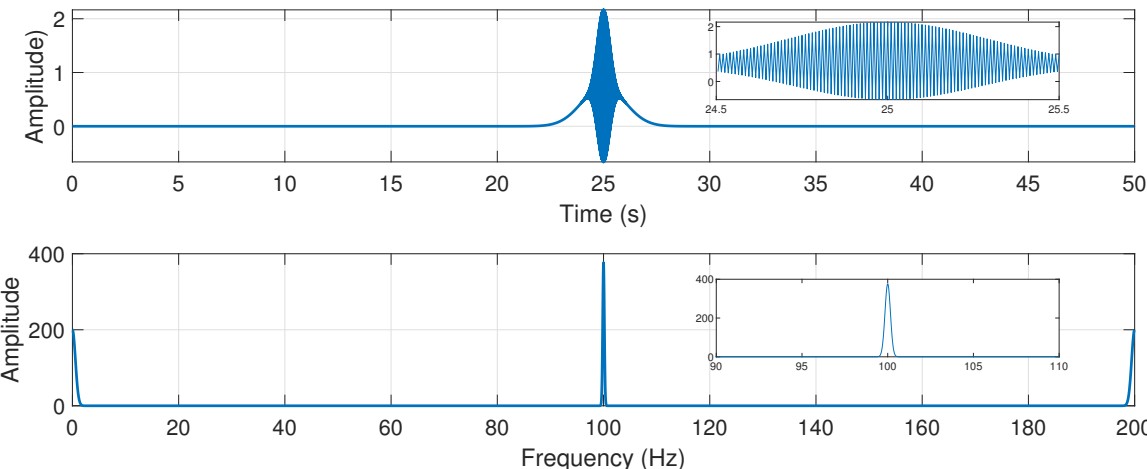

**Figure 4.** Multi-band window (15) in the time domain (top) and in the frequency domain (bottom). This window contains 2 Gaussian windows, in 2 adjacent frequency bands of 100 Hz, one with $\alpha = 1$ and the other one with $\alpha = 12.6$.

2.  The spectrogram of the current signal (17) is built, using the multi-band window (18) as sliding window (Figure 5). It shows two elementary spectrograms in adjacent TF regions, obtained with a single run of the STFT algorithm. The bottom one ($\alpha = 1$) locates the sinusoidal component at 50 Hz but blurs the chirp component. On the contrary, the top one ($\alpha = 12.6$) locates the chirp component but widens the sinusoidal component. This second Gaussian window, and the spectrogram that it generates, have been shifted to the frequency band [100–200 Hz].

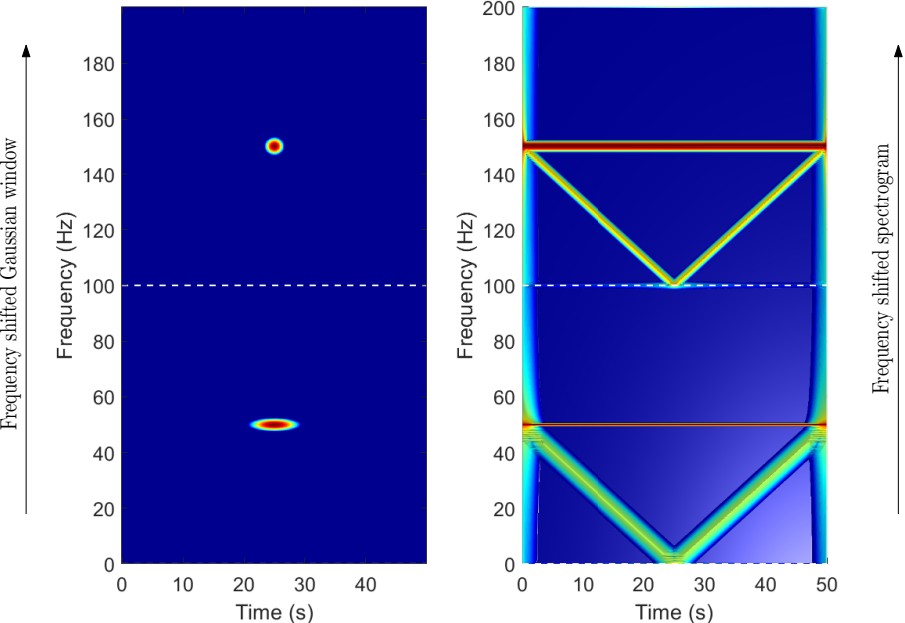

**Figure 5.** Spectrogram of the current signal (17) obtained in step 2 with the multi-band window (right). The Gaussian component of the window with $\alpha = 1$ (bottom, left) locates the sinusoidal component at 50 Hz (bottom, right) but fails to resolve the chirp component. On the contrary, the Gaussian component with $\alpha = 12.6$ (top, left) locates the chirp component but widens the sinusoidal component (top, right). The Gaussian window that is frequency shifted (left) generates a spectrogram that is also frequency shifted (right).

3.  All the stacked, elementary spectrograms obtained in step 2 are shifted back to the frequency band [0–$f_b$], as shown in Figure 6. This process has a negligible computational cost, just the renumbering of the frequency axis of each elementary spectrogram.

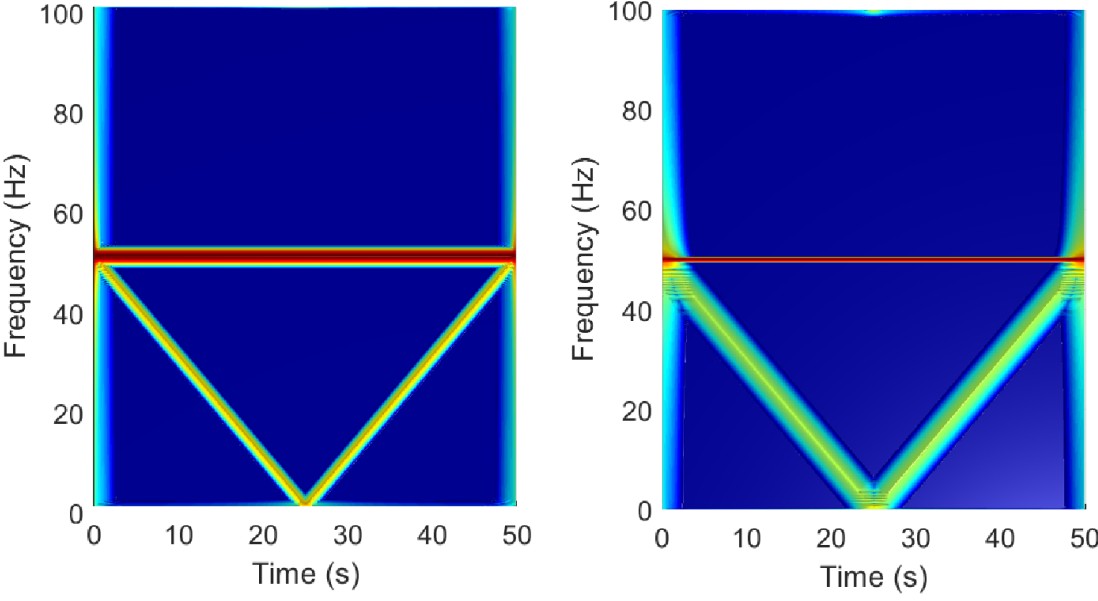

**Figure 6.** Relocation in the frequency axis of the elementary spectrograms, so that all of them span the same frequency band [0–$f_b$].

4.  The points with the same time-frequency coordinates in all the relocated spectrograms obtained in step 3 (Figure 6) are combined to give a single high resolution spectrogram of the TF region of diagnostic interest, in the frequency band [0–$f_b$] (Figure 7). The combination process used in this work consists in selecting, for each point of this region, the minimum value obtained among all the relocated spectrograms. The final result shows with a high resolution both the sinusoidal component at 50 Hz and the chirp component.

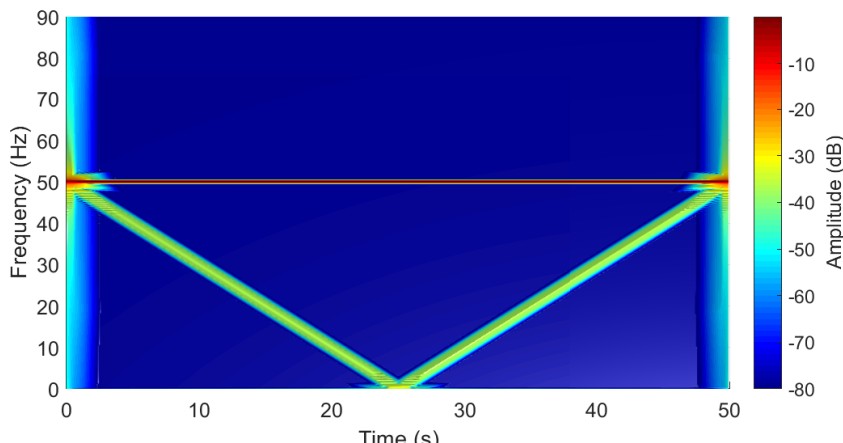

**Figure 7.** High resolution spectrogram of the current signal (17) —for each point of the time frequency (TF) region of interest, the minimum value obtained among all the stacked spectrograms of Figure 5, right, is selected. The final result shows with a high resolution both the sinusoidal component at 50 Hz and the chirp component.

## 4. Experimental Validation

For the experimental verification of the proposed approach, an IM whose characteristics are given in Appendix A has been prepared with a forced rotor fault, by drilling a hole in one of the rotor bars. The stator current during a startup transient has been acquired with a frequency rate $f_s = 5000$ Hz, during 12 s, using a current probe whose characteristics are given in Appendix B and it is shown in Figure 8.

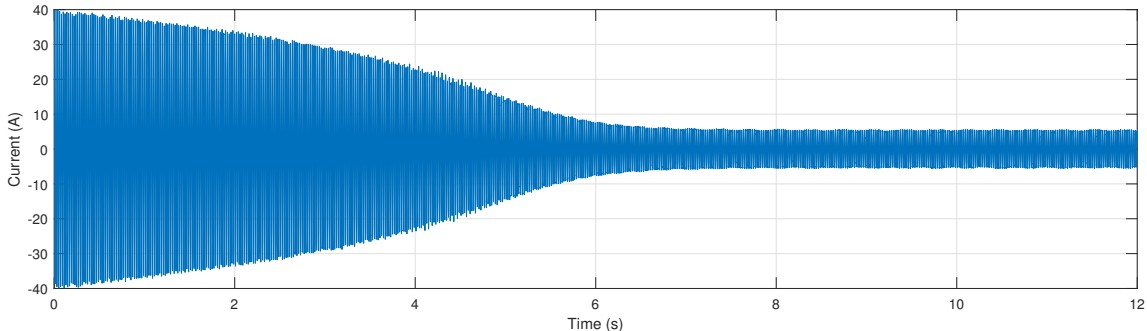

**Figure 8.** Start-up current of the motor of Appendix A with a broken bar fault.

From Table 1, the characteristic frequencies of the fault harmonics in an IM with a rotor asymmetry, such as a broken bar, are, for the main fault harmonics ($k/p = 1$ in Table 1),

$$f_{bb} = f_1(1 \pm 2s) \tag{19}$$

In particular, the fault harmonic with a frequency given by

$$f_{LSH} = f_1(1 - 2s) \tag{20}$$

which is known as the lower side-band harmonic (LSH), is commonly tracked for the diagnosis of rotor asymmetries. During a start-up transient, the trajectory of the LSH in the TF plane given by (20) generates a typical V-shaped fault signature [46], with a frequency that initially ($s = 1$) is equal to the fundamental frequency $f_1$, decreases to 0 ($s = 0.5$) and then increases again until its steady-state regime value of $f_1(1 - 2s) \approx f_1$, according to (20). The ability to detect this fault harmonic using the proposed method is to be assessed in this section.

The three steps presented in Section 3.1 will be followed in this experimental case to obtain the spectrogram of the current signal presented in Figure 8.

1.  The analysis window (15) is built:

    *   first, the bandwidth of diagnostic interest is established. In this case, from (20), the frequency band of interest is the [0–50 Hz] band. Due to the presence of higher order harmonics in the current spectrum, apart form the LSH, a wider band [0–125 Hz] has been selected, in order to better assess the strength of the LSH compared with them. In this way, with a sampling frequency of 5000 Hz, the maximum number of elementary Gaussian windows from (14) is $Ng = 5000/125 = 40$ windows.
    *   second, the parameter $\alpha_k$ of each of these windows in (15) is selected. In this case, a linear range of the parameter $\alpha$ is used, covering the range [$\alpha_{min} = 1 - \alpha_{max} = 700$].

    The resulting multi-band window, applying (15), is

$$gm(t) = \sum_{k=0}^{39} \left(\frac{k\alpha_k}{\pi}\right)^{1/4} e^{-\frac{k\alpha_k}{2}t^2} e^{jk2\pi 125} \tag{21}$$

with

$$\alpha_k = 1 + k\frac{700 - 1}{39} \quad k = 0, 1, \ldots, 39 \tag{22}$$

The multi-band window (21) is displayed in Figure 9, which shows the real (Figure 9 top) and the imaginary part (Figure 9 middle) of the window, as well as its spectrum (Figure 9 bottom).

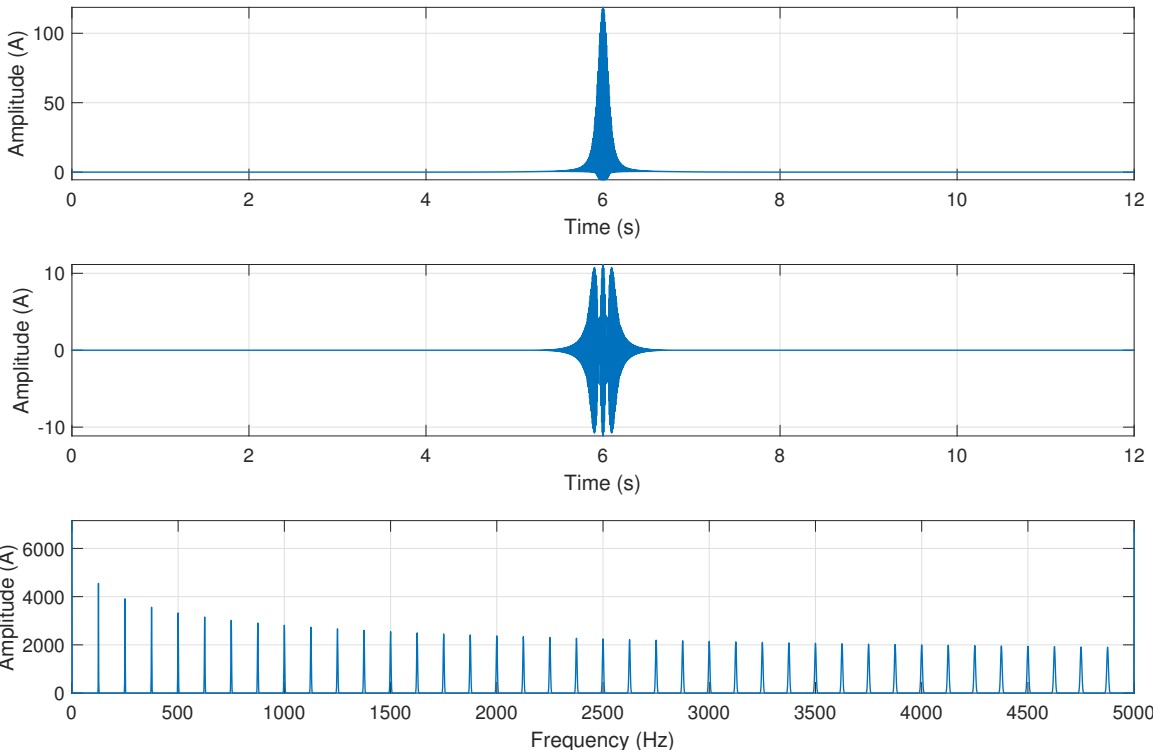

**Figure 9.** Multi-band window (21) in the time domain (real part, **top**, imaginary part, **middle**) and in the frequency domain (**bottom**). This window contains 40 Gaussian windows, located in 40 adjacent frequency bands of 125 Hz, with values of $\alpha$ ranging from $\alpha_{min} = 1$ to $\alpha_{max} = 700$.

2. The spectrogram of the current signal of Figure 8 is built. First, the current signal is low-pass filtered, keeping only the frequency bins of its spectrum lower than 125 Hz. After, and using the multi-band window (21) as sliding window, the STFT algorithm (4) is applied, which generates the spectrogram shown in Figure 10. This spectrogram contains 40 elementary spectrograms in adjacent TF regions (Figure 10, right), obtained with 40 different Gaussian windows (Figure 10, left), at the cost of a single run of the STFT algorithm (6 seconds with the computer of Appendix C). This computation time is not compatible with real-time applications but this is not important for the diagnosis of faults that develop progressively along hours, days or months, such as rotor asymmetries, eccentricities or bearing faults.

Two of the 40 individual Gaussian windows shown in Figure 10 are displayed in Figure 11, left, along with their corresponding current spectra (Figure 11, right). The two zoomed bands corresponds to the spectrogram located in the base frequency band [0–125 Hz] (Figure 11, bottom), which defines clearly the fundamental component but blurs the fault harmonics, and to the spectrogram shifted to the frequency band [1500–1625 Hz] (Figure 11, top), which defines clearly the fault harmonics but widens the fundamental component.

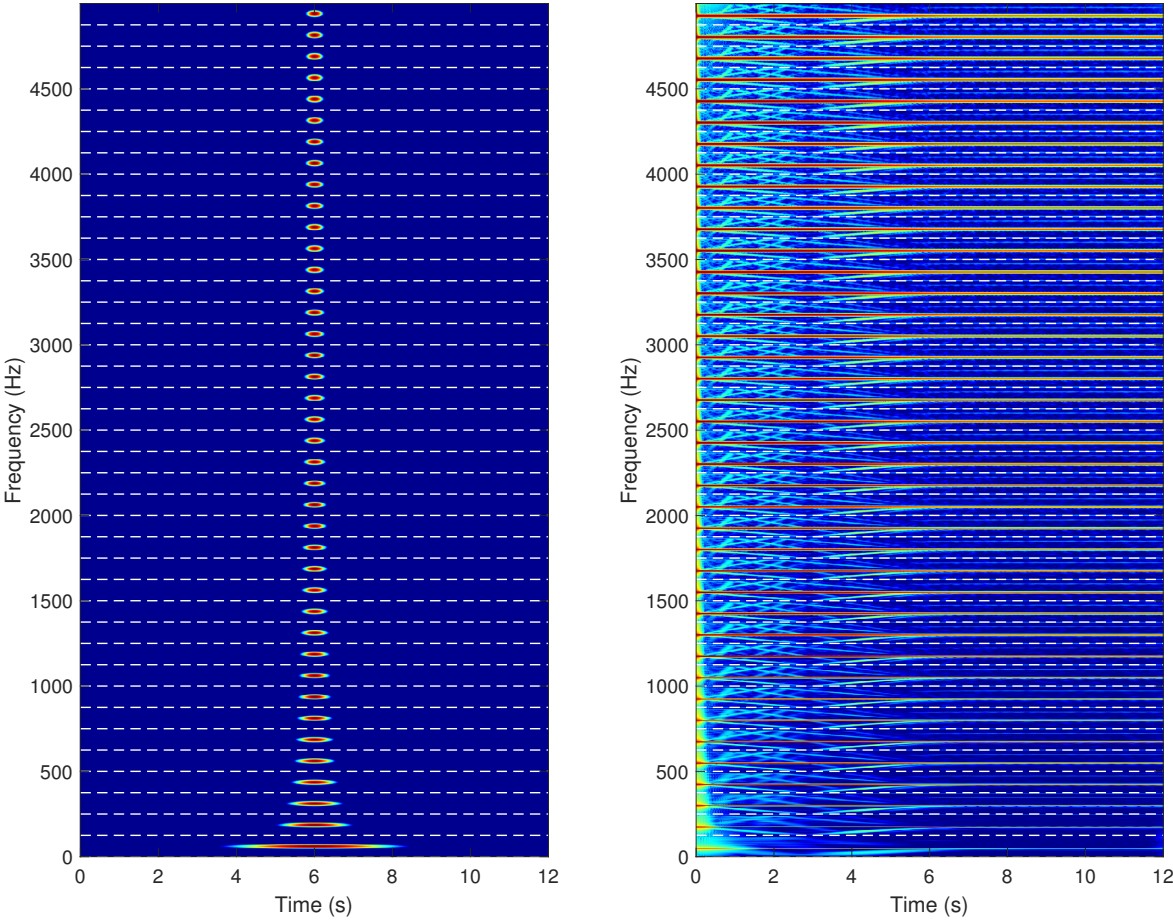

**Figure 10.** Spectrograms of the current of Figure 8 (**right**) obtained with the multi-band window (21) (**left**). The 40 stacked spectrograms have been obtained with a single run of the short-time Fourier transform (STFT) algorithm.

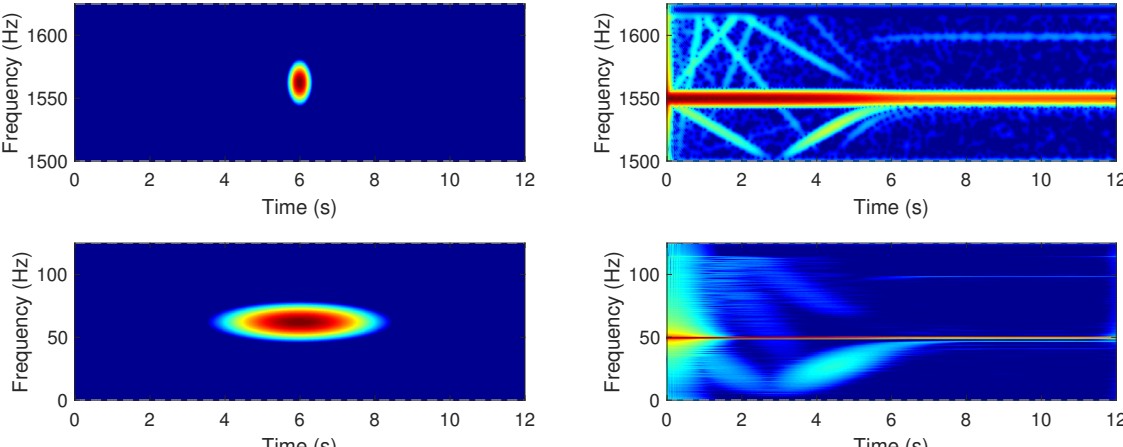

**Figure 11.** Zoom of Figure 10 showing two of the individual Gaussian windows contained in the multi-band window of Figure 10 (**left**) and the corresponding spectrograms generated with them (**right**). The two zoomed bands corresponds to the spectrogram located in base frequency band [0–125 Hz] (**bottom**), which defines clearly the fundamental component but blurs the fault harmonics and to the spectrogram shifted to the frequency band [1500–1625 Hz] (**top**), which defines clearly the fault harmonics but widens the fundamental component.

3. All the stacked, elementary spectrograms obtained in step 2 (Figure 10, right), are relocated in the base frequency band [0–125 Hz], by renumbering their frequency axis.

4. All the relocated, elementary spectrograms obtained in step 3 (Figure 10, right), are combined to give a high resolution spectrogram of the TF region of diagnostic interest (Figure 12). The combination process used in this work consists in selecting, for each point of this [0–125 Hz] region, the minimum value obtained among all the relocated spectrograms. The final result shows with a high resolution both the sinusoidal component at 50 Hz and the LSH fault component. Unlike the individual spectrograms, the optimized spectrogram clearly shows the LSH not only during the transient period but also when the steady state is reached; it is also remarkable the set of fault-related second-order components that are revealed, which helps to get a more reliable diagnostic.

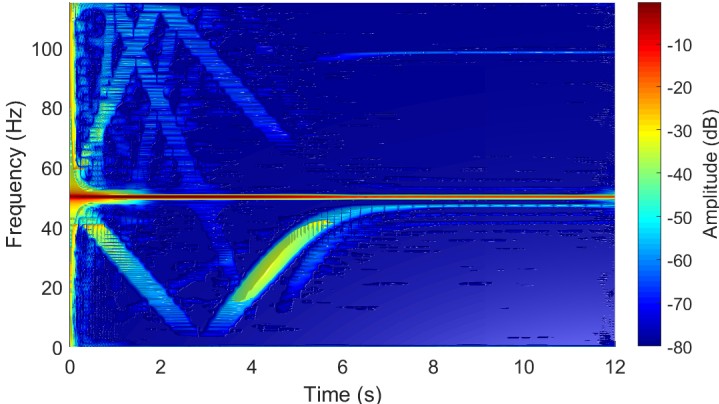

**Figure 12.** High resolution spectrogram of the current of Figure 8—for each point of the TF region of interest, the minimum value obtained among all the stacked spectrograms of Figure 10, right, is selected. The final result shows with a high resolution both the sinusoidal component at 50 Hz and the lower side band harmonic (LSH) fault component.

## 5. Conclusions

In this work, a novel technique for performing the fault diagnosis of IMs working in transient regime has been presented and validated experimentally. This technique consists in building a multi-band analysing window, composed by several, different Gaussian windows, stacked in the frequency domain. When this window is multiplied by the current signal in the time domain, it generates a spectrum which contains all the spectra generated by each of the Gaussian windows that form the multi-band window. In this way, a spectrogram containing even hundreds of different Gaussian windows can be obtained at the cost of a single run of the STFT algorithm. The selection of the parameters of the individual windows that compose the multi-band window can be setup by the user using different criteria, without affecting the performance of the proposed approach. In this work, a blind approach has been used, by selecting a range of individual windows that cover a wide range of, a priori, unkown fault harmonic components. This approach can be particularly useful for developing automated AI-based diagnostic systems, which can operate without expert assistance, and for the detection of different types of faults. In this case, a single multi-band window can avoid the design of multiple analysis windows specifically designed for each single type of fault.

**Author Contributions:** R.P.-P. and A.S.-B. developed the theoretical explanation of the method. J.B.-V. and J.M.-R. designed and carried out the experimental validations; M.P.-S. wrote the paper and M.R.-G. carried out the revision the paper.

**Funding:** This research was funded by the Spanish "Ministerio de Ciencia, Innovación y Universidades (MCIU)", the "Agencia Estatal de Investigación (AEI)" and the "Fondo Europeo de Desarrollo Regional (FEDER)" in the framework of the "Proyectos I+D+i - Retos Investigación 2018", project reference RTI2018-102175-B-I00 (MCIU/AEI/FEDER, UE).

**Conflicts of Interest:** The authors declare no conflict of interest.

## Appendix A. Motor Characteristics

Three-phase induction machine. Rated characteristics: $P = 1.1$ kW, $f = 50$ Hz, $U = 230/400$ V, $I = 2.7/4.6$ A, $n = 1410$ r/min , $\cos \varphi = 0.8$.

## Appendix B. Current Clamp

Chauvin Arnoux MN60, Nominal measuring scope: 100 mA–20A, ratio input/output: 1 A/100 mV, intrinsic error: $\leq$ 2% + 50 mV, frequency use: 400 Hz–10 kHz.

## Appendix C. Computer Features

CPU: Intel Core i7-2600K CPU @ 3.40 GHZ RAM memory: 16 GB, Matlab Version: 9.6.0.1072779 (R2019a).

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
