# Peer review of "Multi-Band Frequency Window for Time-Frequency Fault Diagnosis of Induction Machines"

_energies, doi:10.3390/en12173361_

Round 1

Reviewer 1 Report

see attachment

requesting a few additional (mathematical) references!

Reviewer 2 Report

I enclose the pdf where you can read the comments related to yellow highlighted parts of the paper.

Furthermore, I submit to the authors the following considerations:

a) In the paper, authors report a relatively high PC computational time (6 s) that is not likely compatible with a real time application: they should clarify the possible implementation of the proposed technique.

b) Authors claim that "this approach can be particularly useful for automated diagnostic system, which can operate without expert assistance". In my opinion, taking into account that failures may have several differences as for typology and intensity, not to to mention that they can be combined together, the pattern recognition in spectrograms probably is a very sophisticated task. Authors should give some hint on the roadmap for the practical application of this technique.

c) What is the authors' opinion as for the method robustness with respect to environmental disturbances, for instance small supply voltage unbalance/asymmetry that could be misinterpreted as failures or mask real failures?
